# Oxygenation Kinetics of Three Quadriceps Muscles During Squatting Exercise in Trained Men

**DOI:** 10.3390/sports12100283

**Published:** 2024-10-17

**Authors:** Konstantinos Mavridis, Anatoli Petridou, Athanasios Chatzinikolaou, Vassilis Mougios

**Affiliations:** 1Laboratory of Evaluation of Human Biological Performance, School of Physical Education and Sport Science at Thessaloniki, Aristotle University of Thessaloniki, 54124 Thessaloniki, Greece; apet@phed.auth.gr (A.P.); mougios@auth.gr (V.M.); 2Laboratory of Physical Performance, School of Physical Education and Sport Science at Komotini, Democritus University of Thrace, 69150 Komotini, Greece; achatzin@phyed.duth.gr

**Keywords:** NIRS, rectus femoris, SmO_2_, squat, tHb, vastus lateralis, vastus medialis

## Abstract

This study aimed to monitor the oxygenation and blood supply in three quadriceps muscles [the vastus lateralis (VL), vastus medialis (VM), and rectus femoris (RF)] during squatting exercise to exhaustion. Eighteen young resistance-trained males performed five sets of 15 back squats in a Smith machine, with two warm-up sets [at 14% and 45% of the 15-repetition maximum (15RM)] and three main sets at 100% of the 15RM. Three near-infrared spectroscopy devices were attached to the VL, VM, and RF to record the muscle oxygen saturation (SmO_2_) and total hemoglobin (tHb, an index of muscle blood supply). The blood lactate concentration was measured after each set with a portable analyzer. The SmO_2_ and tHb data were analyzed by repeated-measures two-way ANOVA (muscle × set). Lactate data were analyzed by repeated-measures one-way ANOVA. The statistical significance was set at α = 0.05. The SmO_2_ dropped during each set (hitting zero in many instances) and was reinstated during recovery. The three main sets caused severe deoxygenation in the VL and VM, as opposed to moderate deoxygenation in the RF. From one set to the next, the initial value and the drop in the SmO_2_ increased, whereas the final SmO_2_ value decreased. The tHb increased in the VL, did not change considerably in the VM, and decreased in the RF during each set. The blood lactate concentration increased gradually from one set to the next, reaching about 10 mmol/L. These findings show pronounced differences in the physiological and metabolic responses of three quadriceps muscles to squatting exercise, thus highlighting the importance of studying such responses at multiple sites.

## 1. Introduction

Near-infrared spectroscopy (NIRS) is an analytical method that can be used non-invasively to evaluate local changes in the muscle oxygenation and blood supply. This is achieved by taking advantage of the differences in the absorption spectra of oxygenated and deoxygenated heme (the prosthetic group of hemoglobin, myoglobin, and cytochromes). The main variable that a NIRS device measures when applied to a muscle is muscle oxygen saturation (SmO_2_, defined as oxygenated heme/total heme × 100), a measure of muscle oxygenation. Another variable is the total heme, which has been used to evaluate changes in the muscle blood volume and muscle blood supply [1,2]. NIRS has been extensively used to evaluate the balance between muscle oxygen delivery and utilization in real time under various conditions, including exercise [3], with more studies having been performed on endurance rather than resistance exercise [4].

The squat is a fundamental group of exercises that all-level athletes and non-athletes can use to improve performance and health. To our knowledge, only four published studies have examined muscle oxygenation in squat variations [5,6,7,8]; all of these studies focused on the vastus lateralis (VL). Three studies have examined other quadriceps muscles, that is, the vastus medialis (VM) and rectus femoris (RF), separately during dynamic resistance exercises other than squats [9,10,11]. Two studies have examined the oxygenation of more than one muscle simultaneously: the VL and RF in knee extension [9], and the biceps brachii and brachioradialis in elbow flexion [12]. Thus, there seems to be a shortage of information on the oxygenation of multiple muscles participating in a specific exercise. This is important, since different muscles may have different activation and participation patterns in a given movement.

Therefore, examining the oxygenation of multiple muscles may offer a more complete picture of the exercise’s local metabolic burden and contribute a useful tool to scheduling resistance training, as is the case with endurance training, where the deoxyhemoglobin breakpoint and training zones based on the SmO_2_ have been proposed as indices for setting training intensities [13,14]. Establishing similar breakpoints or zones in resistance training could result in more accurate scheduling and more targeted adaptations to training.

Thus, the present study aimed to monitor the oxygenation and blood supply in three quadriceps muscles (the VL, VM, and RF) during a five-set protocol of back squats in a Smith machine. It was hypothesized that the three muscles would differ concerning these variables. To better evaluate the muscle oxygenation and blood supply responses to this exercise, we measured additional biochemical and physiological variables. These were the heart rate (HR) and blood pressure (as indices of the systemic oxygen supply), as well as blood lactate (an index of the anaerobic energy supply). It was further hypothesized that these parameters would have positive relationships with local oxygen utilization in response to increased energy demands by exercising muscles.

## 2. Materials and Methods

### 2.1. Study Sample and Ethics

Eighteen young, trained, healthy men (students at the School of Physical Education and Sport Science) participated voluntarily in this experimental, acute, single-arm study. This was a convenience sample, and its size was determined by an a priori power analysis using the G*Power software (version 3.1.9.2, Kiel University, Kiel, Germany), which showed that 16 individuals were needed to detect a significant effect with α at 0.05, a power of 0.8, a medium effect size (partial η^2^ of 0.1 in an analysis of variance, ANOVA), and a correlation coefficient of 0.5 between repeated measures. Inclusion criteria were (i) experience in resistance exercise and, specifically, squat in a Smith machine, (ii) thickness of subcutaneous fat below 12 mm at the sites of application of the NIRS devices (as dictated by the manufacturer for valid recordings), and (iii) the absence of any musculoskeletal injury in the lower limbs. The volunteers signed an informed consent form before participating in the study. The study was approved by the Ethics Committee of the School of Physical Education and Sport Science at Thessaloniki (approval number 112/28-3-2022) and was performed according to the ethical standards of the Helsinki Declaration.

### 2.2. Study Design

The participants visited the laboratory twice between 9:00 and 12:00, three to seven days apart. On the day preceding each visit, they were reminded to arrive well-rested, having had a light breakfast up to two hours before. During the first, preparatory visit, the following measurements were performed on each participant: skinfold thickness at the VL, VM, and RF (to ascertain that the subcutaneous fat layer did not exceed 12 mm), body mass, height, and the 15-repetition maximum (15RM) in the back squat in a Smith machine through repeated sets with gradually increasing loads.

During their second, main visit, the participants performed a five-set protocol of back squats consisting of two warm-up sets (one with the bar only and the other at 45% of the 15RM) and three main sets at 100% of the 15RM. During the exercise protocol, the SmO_2_, total heme, and HR were continuously monitored. Additionally, blood pressure and blood lactate were measured between sets.

### 2.3. Testing Protocol

The testing protocol during the main visit is depicted in Figure 1. Initially, three NIRS monitors (Moxy, Idiag, Fehraltorf, Switzerland) were affixed to the most prominent points of the VL, VM, and RF muscle bellies on each participant’s dominant leg using adhesive tape and elastic bandages. Data were collected wirelessly through the Moxy software (v.1.0, Moxy, Idiag, Fehraltorf, Switzerland) at a sampling frequency of 2 Hz. In addition to the SmO_2_, total hemoglobin (tHb) was recorded, which is a dimensionless quantity corresponding to the local total (that is, oxygenated and deoxygenated) heme concentration. Because the total amount of heme inside the muscle fibers cannot change during an exercise session, changes in the total heme concentration reflect changes in the amount of hemoglobin inside the blood vessels, hence the term tHb. Thus, tHb can be used to monitor changes in the local muscle blood supply. Also, a wireless HR monitor (Polar, Kempele, Finland) was fastened to each participant’s chest with a belt. The HR was monitored through the Polar Flow application.

Participants then performed the following sets of back squats in a Smith machine until the thighs were parallel to the ground and the bar was in a high position: a warm-up set with the bar only (corresponding, on average, to 14% of the 15RM), a second warm-up set at 45% of the 15RM, and three main sets at 100% of the 15RM. All of the sets contained 15 repetitions and were interspersed with 2 min of rest. For the proper execution of the sets, the participant was guided by a video of a model trainee performing squats to the rhythm of a digital metronome, set at 40 beats per minute (that is, striking every 1.5 s). The participant was asked to start from the standing position at the first strike, reach the squatting position (marked with a horizontal elastic band that the volunteer touched with his buttock) at the second strike while moving as smoothly as possible, return to the standing position at the third strike, and so on until the end of the set, thus keeping the duration of each repetition at 3 s and that of the set at 45 s.

Arterial blood pressure was measured at baseline and as quickly as possible after each set (except for the first one) with an electronic blood pressure monitor (Microlife BP A1 Basic, Widnau, Switzerland). Finally, lactate was measured in capillary blood from a fingertip with a portable analyzer (Lactate Scout 4, EKF Diagnostics, Cardiff, UK) one minute after the end of each set (except for the first one). After the final set, measurements were taken every minute until a value lower than the previous one (declared the peak value) was recorded.

### 2.4. Statistical Analysis

Data are presented as the mean and SD. The normality of data distribution was examined with the Shapiro–Wilk test and was found not to differ significantly from normal. Two-way ANOVA with repeated measures on the muscle and set was used for the analysis of the SmO_2_ and tHb data. One-way ANOVA with repeated measures on the set was used for the analysis of the HR, blood pressure, and blood lactate data. Sphericity was examined with Mauchly’s test and, when violated, the Greenhouse–Geisser correction was used. Significant outcomes were followed up by a simple main effect analysis with Sidak adjustment for multiple comparisons. The change in SmO_2_ (ΔSmO_2_) and tHb (ΔtHb), that is, the difference between the final and initial values in each set and muscle, was compared to zero through a one-sample Student’s *t* test. Variables were correlated by Pearson’s correlation analysis.

The level of statistical significance was set at α = 0.05. Effect sizes (ESs) were determined as partial η^2^ and were classified as small (0.01–0.058), medium (0.059–0.137), or large (>0.137) according to Cohen [15]. SPSS, version 28.0 (SPSS, Chicago, IL, USA), was used for all analyses.

## 3. Results

### 3.1. Characteristics of the Participants

Participants were 22.2 ± 2.4 years old and had training experience of 5.4 ± 3.6 years, a body mass of 75.2 ± 4.5 kg, a height of 1.76 ± 0.05 m, and a body mass index of 24.2 ± 2.0 kg/m^2^. The resting HR was 79 ± 12 bpm, the resting systolic pressure was 123 ± 12 mmHg, and the resting diastolic pressure was 73 ± 8 mmHg. The 15RM was 88 ± 23 kg. The 1RM was calculated to be 135 ± 36 kg, assuming the 15RM to be 65% of the 1RM [16]. The 1RM/body mass was 1.80 ± 0.49.

### 3.2. SmO_2_

Figure 2 illustrates the SmO_2_ kinetics in the three muscles during the entire testing protocol. In most cases, the SmO_2_ declined sharply at the onset of each set, and more gradually afterward. Similarly, the SmO_2_ increased rapidly at the beginning of recovery, and more gradually afterward, returning to baseline or to even higher values before the onset of the next set.

Figure 3 presents the initial SmO_2_, final SmO_2_, and ΔSmO_2_ in each set and muscle as boxplots for the better visualization of the interindividual variability. We found significant main effects for both muscle and set on all three variables (*p* ≤ 0.001; ES = 0.328–0.690 for the main effect of the muscle and 0.533–0.799 for the main effect of the set). Also, there was a significant interaction of muscle and set in the initial and final values (*p* < 0.001; ES = 0.436 and 0.319, respectively), as well as a marginally significant interaction in ΔSmO_2_ (*p* = 0.072; ES = 0.119). The initial SmO_2_ values (Figure 3A), regardless of the set, were the highest in the RF (81.1 ± 8.5%), followed by the VM (79.3 ± 6.7%) and VL (74.0 ± 6.0%).

The initial SmO_2_, regardless of the muscle, was 67.9 ± 9.4, 77.9 ± 6.4, 80.1 ± 5.5, 82.5 ± 4.7, and 82.3 ± 5.0% in sequence in the five sets. Pairwise comparisons showed that the VL differed significantly from the VM and RF in the initial SmO_2_ of most sets, and that there was a significant gradual increase from one set to the next in the VL and VM, which was only partially seen in the RF.

In terms of the final SmO_2_ (Figure 3B), the VL displayed the lowest values (8.6 ± 6.7%), followed by the VM (16.1 ± 6.8%) and RF (44.8 ± 21.5%). Aside from the highest values, the RF exhibited the highest interindividual variability. The final SmO_2_, regardless of the muscle, was 33.0 ± 12.3, 27.1 ± 10.8, 20.3 ± 10.2, 17.9 ± 9.1, and 17.4 ± 7.5% in sequence in the five sets. Notably, the final SmO_2_ reached zero for some participants in the VL (all sets), VM (two sets), and RF (three sets). All three muscles differed significantly from each other in most sets. In all muscles, there was a trend for a gradual decrease in the final SmO_2_ from one set to the next.

The drop in the SmO_2_ within the sets (Figure 3C) was the highest in the VL (65.5 ± 7.0%), followed by the VM (63.2 ± 8.0%) and RF (36.2 ± 21.2%). This drop, regardless of the muscle, was 34.8 ± 13.8, 53.5 ± 10.1, 59.8 ± 8.6, 64.6 ± 9.0, and 64.8 ± 8.0% in sequence in the five sets. The VL and VM did not differ significantly in any set regarding the ΔSmO_2_, although both differed from the RF in all sets. Within each muscle, most sets differed significantly from each other.

### 3.3. tHb

Figure 4 illustrates the tHb kinetics in the three muscles during the entire testing protocol. In most cases, tHb exhibited a decline at the beginning of the set, followed by an increase in the VL, a relative stabilization in the VM, and a decrease in the RF during the remainder of the set. During the rest intervals, tHb remained relatively stable in the VL, whereas it increased in the VM and RF.

Figure 5 presents the initial tHb, final tHb, and ΔtHb in each set and muscle. We found significant main effects for both the muscles and sets, as well as a significant interaction of muscles and sets, in all three variables (*p* ≤ 0.05; ES = 0.427–0.747 for the main effect of the muscle and 0.255–0.643 for the main effect of the set, and 0.124–0.252 for the interaction).

The initial tHb values (Figure 5A), regardless of the set, were the highest in the VM (12.54 ± 0.27) and VL (12.51 ± 0.24), followed by the RF (12.20 ± 0.34). The initial tHb, regardless of the muscle, was 12.32 ± 0.25, 12.39 ± 0.22, 12.41 ± 0.22, 12.47 ± 0.22, and 12.49 ± 0.22 in sequence in the five sets. Pairwise comparisons showed that the RF differed significantly from the VL and VM in the initial tHb of all sets, and that there was a significant gradual increase from one set to the next in the VL and VM, which was only partially seen in the RF.

In terms of the final tHb (Figure 5B), the RF displayed the lowest values (11.82 ± 0.39), followed by the VM (12.52 ± 0.51) and VL (12.72 ± 0.38). The final tHb, regardless of the muscle, was 12.36 ± 0.52, 12.35 ± 0.54, 12.34 ± 0.61, 12.40 ± 0.56, and 12.38 ± 0.56 in sequence in the five sets. Pairwise comparisons revealed that the RF differed significantly from the VL and VM in the final tHb of all sets, with only one significant difference between sets in a muscle (specifically, the VL).

The ΔtHb (Figure 5C) was significantly higher than zero in all sets in the VL (*p* < 0.01), averaging 0.18 ± 0.19, which was not significantly different from zero in any set in the VM (*p* > 0.2), averaging −0.02 ± 0.35, and significantly lower than zero in all sets in the RF (*p* < 0.05), averaging −0.30 ± 0.30. The ΔtHb, regardless of the muscle, was 0.04 ± 0.29, −0.04 ± 0.35, −0.07 ± 0.41, −0.08 ± 0.34, and −0.12 ± 0.34 in sequence in the five sets. The RF differed significantly from the VL and VM in all sets. There were only two significant differences between sets in a muscle (specifically, in the RF).

### 3.4. Heart Rate and Blood Pressure

The kinetics of the HR during the five sets and recovery intervals are presented in Figure 6, which shows an increase within each set and a decrease during recovery. The mean HR in the five sets was 115 ± 17, 120 ± 14, 136 ± 15, 135 ± 14, and 136 ± 16 bpm in sequence, with a significant main effect of the set (*p* < 0.001; ES = 0.581) and significant pairwise differences between the first set and the three main sets (all *p* < 0.001).

Blood pressure values (obtained, on average, 37 s after the end of the second, third, fourth, and fifth sets) were, in sequence, 146 ± 15, 146 ± 17, 144 ± 17, and 133 ± 18 mm Hg for systolic and 79 ± 8, 78 ± 8, 78 ± 8, and 75 ± 10 mm Hg for diastolic pressure. Only the systolic pressure values exhibited a main effect of time (*p* = 0.013; ES = 0.226), with the fifth set differing significantly from the second and third sets (*p* < 0.05).

### 3.5. Blood Lactate

The blood lactate concentration increased gradually after sets 2 to 5 (*p* < 0.001, Figure 7). The simple main effect analysis showed significant differences between all sets (*p* < 0.001; ES = 0.794). The blood lactate concentration was 2.2 ± 0.8, 5.4 ± 2.1, 7.5 ± 3.0, and 9.9 ± 3.9 mmol/L in sequence.

### 3.6. Observed Statistical Power

The average observed power of the analyses described above was 0.906, in accordance with the a priori calculation described in Section 2.1.

### 3.7. Correlations

Pearson’s correlation analysis revealed only a few scattered correlations between the SmO_2_, tHb, mean HR, blood pressure, and lactate variables.

## 4. Discussion

The present study examined, for the first time, the differences in the oxygenation and blood supply between three quadriceps muscles during five squatting exercise sets of differing loads in trained men. Our main findings are the remarkably different SmO_2_ and tHb kinetics between the three muscles.

The SmO_2_ at the beginning of each set, although statistically different between muscles in most cases, displayed the same upward parabolic trend from one set to the next within each muscle (see Figure 2 and Figure 3A). This trend shows that the time (2 min) allocated to recovery between sets was more than sufficient for full muscle reoxygenation (even after the maximal sets) and testifies to the efficiency of the cardiovascular system in rapidly restoring the oxidative capacity of muscle after maximal exercise. Additionally, the fact that the largest increases in the initial SmO_2_ were seen after the two warm-up sets attests to the usefulness of warming up before maximal resistance exercise. The overshoot of the SmO_2_ during recovery from resistance exercise has also been described by others in the VL [6,7,17].

The SmO_2_ at the end of each set displayed more pronounced differences between the muscles, with all possible pairwise comparisons but one (the VL vs. VM in the fifth set) yielding significant outcomes, and with the RF displaying considerably higher values than the VL and VM (see Figure 3B). Such a comparison between muscles has not been reported in the literature and suggests a lower (at least aerobic) toll of squatting exercise on the RF compared with the VL and VM. As for the differences between sets, the gradual decrease in the final SmO_2_ can be attributed to the increase in the load from the warm-up to the main sets.

The smaller drop in the SmO_2_ from the beginning to the end of each set in the RF compared with the other two muscles (see Figure 3C), being primarily a result of the higher final SmO_2_ values in that muscle, reinforces the conclusion (mentioned above) for its lower participation in squatting exercise. Similarly, in isokinetic knee extension, the drop in the SmO_2_ was higher in the VL compared with the RF [9]. Differences in the ΔSmO_2_ between sets, located mainly between the two warm-up sets and between the warm-up and main sets (with the drop in the SmO_2_ increasing from the first to the second warm-up sets, and from that to the first main set, while almost stabilizing afterward) agree with the differences in the load (14, 45, and 100% of the 15RM). The small increase from one main set to the next (despite the same load) may be attributed to the incomplete regeneration of phosphocreatine during the two minutes of recovery between sets and, hence, to the gradually higher reliance on aerobic energy production [18].

The higher deoxygenation of the VL and VM than the RF during the squatting exercise may be due to differences in anatomy and biomechanics. Specifically, in squatting exercise, the RF acts on minimal leverage because it is the only head of the quadriceps muscle that has its proximal attachment on the anterior inferior iliac spine of the hip bone, thus starting above the femur [19]. Another possibility (probably related to the previous one) is the different activation of the three muscles. In agreement with this hypothesis, Yavuz and Erdag [20] have reported electromyographic activity values during back squat in the order of VM > VL > RF. However, da Silva et al. [21] have reported values in the order of RF > VL > VM, while Coratella et al. [22] have reported values in the order of VM > VL ≅ RF (all referring to squatting exercise). Notably, all of these findings refer to numerical values, and none of the studies performed statistical comparisons between the muscles. A limitation of the present study is that we did not include electromyographic activity measurements to test the hypothesis that differences in the SmO_2_ between the muscles are due to differences in their activation during the exercise protocol used.

Another possible explanation for the differences in the deoxygenation between the muscles is the difference in the fiber type composition, in the sense that type I fibers are more oxidative, use more oxygen, and have a higher capillarization than type II fibers [18]. Data comparing all three muscles measured in the present study are available only from autopsies [23], and show type I fiber content in the order of VM > RF > VL. Based on this, the differences in oxygen use by the three muscles found in this study cannot be explained by differences in the fiber type composition.

Regarding the tHb, the gradual increase in its initial value from one set to the next is similar to the trend seen in the SmO_2_, and attests to the ability of the cardiovascular system to adjust to the increased demands of the exercising muscles. Differences in the final tHb and ΔtHb between sets in the same muscle were not noteworthy. However, there were remarkable differences between the muscles in all three tHb variables, with the RF having lower values than the VL and VM. This may be due to the lower activation of the RF, as discussed above based on the ΔSmO_2_ results. Only one other study [6] examined the variation in tHb during squatting exercise and found a small increase in the initial tHb, a stabilization in the final tHb, and a small increase in the drop of tHb in the VL during three sets of eight repetitions at 75–80% of the 1RM. These results agree with most of the findings of the present study regarding the VL.

The differences in the HR between the first warm-up set and the three main sets can be explained by the higher load imposed on the exercising muscles and the matching response of the cardiovascular system. The increased blood pressure measured after the sets (compared with rest) agrees with the data of Apkarian [11]. Finally, the gradual increase in blood lactate by 2 to 3 mmol/L from one set to the next indicates a similar and relatively small contribution of the anaerobic carbohydrate breakdown to the energy demands of the exercise protocol. This lactate response is in agreement with the literature [6,24].

## 5. Conclusions

In conclusion, our findings show that three sets of moderate-intensity back squatting exercise to exhaustion caused severe decreases in the oxygen saturation of the VL and VM, as opposed to moderate decreases in the RF. The blood supply increased in the VL, did not change considerably in the VM, and decreased in the RF during each set. Apart from contributing theoretical knowledge on the metabolic burden of squatting exercise on specific quadriceps muscles, the present study may have practical implications in training. Of particular interest, in our view, is the severe hypoxia (with the SmO_2_ reaching zero in several instances, as mentioned in the Results section) elicited specifically in the VL and VM by the moderate-intensity protocol to exhaustion employed in the present study, which may serve as a hypertrophic stimulus [25,26] by increasing the expression of proteins associated with muscle cell differentiation and hypertrophy [27,28], although the exact mechanism remains unclear. Another practical implication is that two minutes of recovery between sets are sufficient for full muscle reoxygenation in similar protocols. In all, our findings reveal pronounced differences between three muscles of the main group participating in squatting exercise (that is, the quadriceps) in their physiological and metabolic responses, thus highlighting the importance of studying such responses at multiple sites in future studies.

## Figures and Tables

**Figure 1 sports-12-00283-f001:**
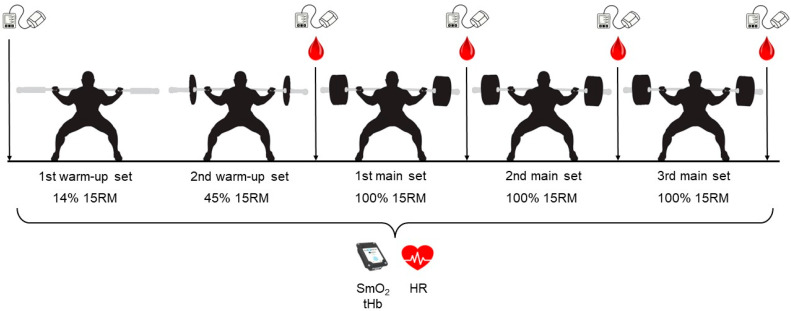
Testing protocol of 5 sets of back squats in a Smith machine. Muscle oxygen saturation (SmO_2_), total hemoglobin (tHb), and heart rate (HR) were monitored during the entire protocol. The blood pressure monitor icon indicates blood pressure measurements.

**Figure 2 sports-12-00283-f002:**
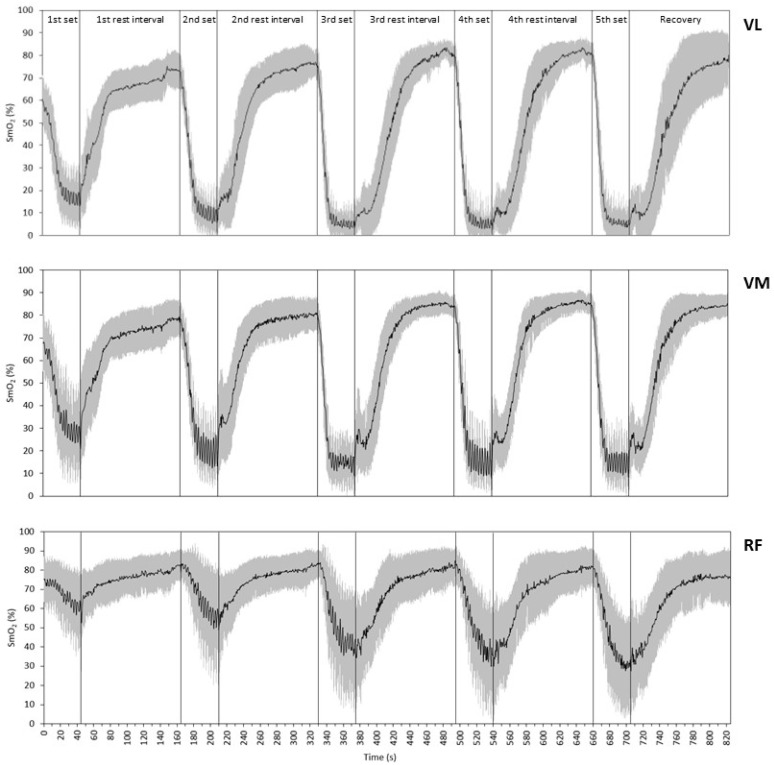
Muscle oxygen saturation (SmO_2_) kinetics in the vastus lateralis (VL), vastus medialis (VM), and rectus femoris (RF) during the testing protocol. Black lines indicate the mean and gray whiskers indicate the SD.

**Figure 3 sports-12-00283-f003:**
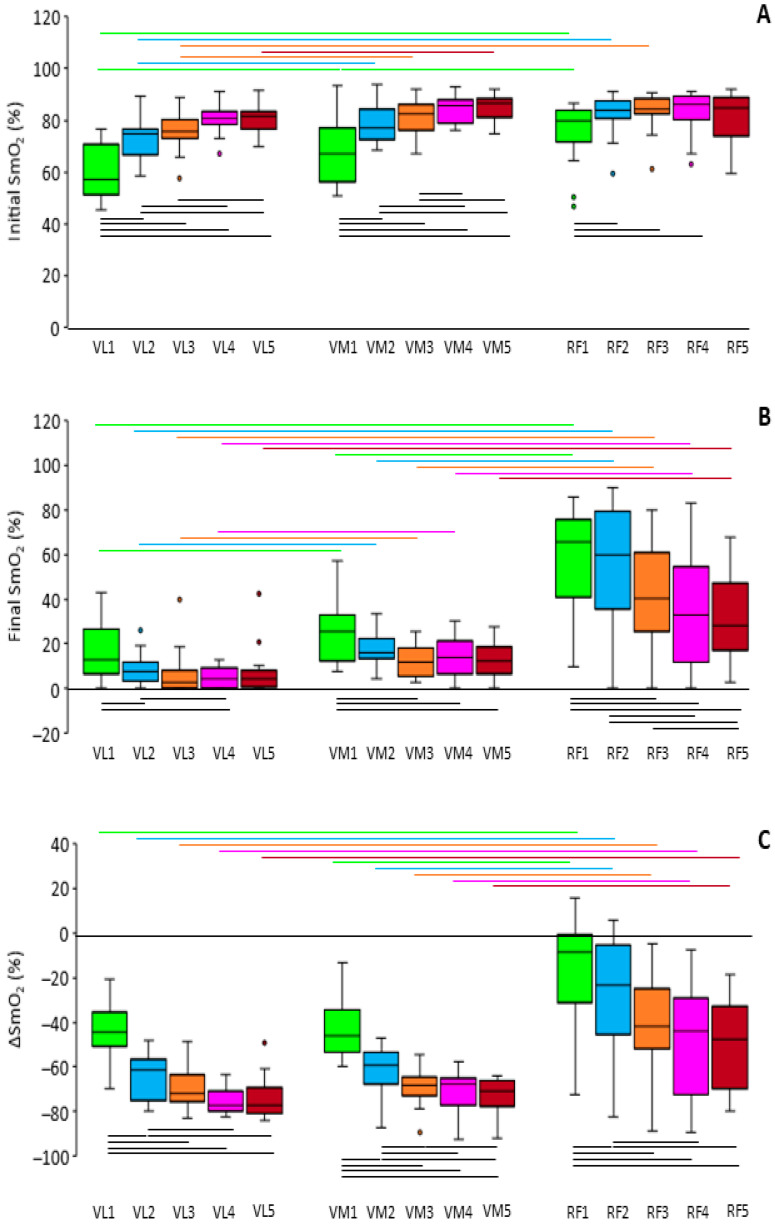
Boxplots of the initial values (**A**), final values (**B**), and change in the SmO_2_ (**C**) in the vastus lateralis (VL), vastus medialis (VM), and rectus femoris (RF) in each of the 5 sets. Each box represents the interquartile range, and the center line represents the median. Whiskers are extended to the most extreme data point, that is, no more than 1.5 times the interquartile range from the edge of the box (Tukey style). Dots represent outliers. Horizontal lines above the boxplots represent significant differences between the muscles in the same set (*p* < 0.05) and have the same color as the set to facilitate the reader. Horizontal lines below the boxplots represent significant differences between the sets in the same muscle (*p* < 0.05).

**Figure 4 sports-12-00283-f004:**
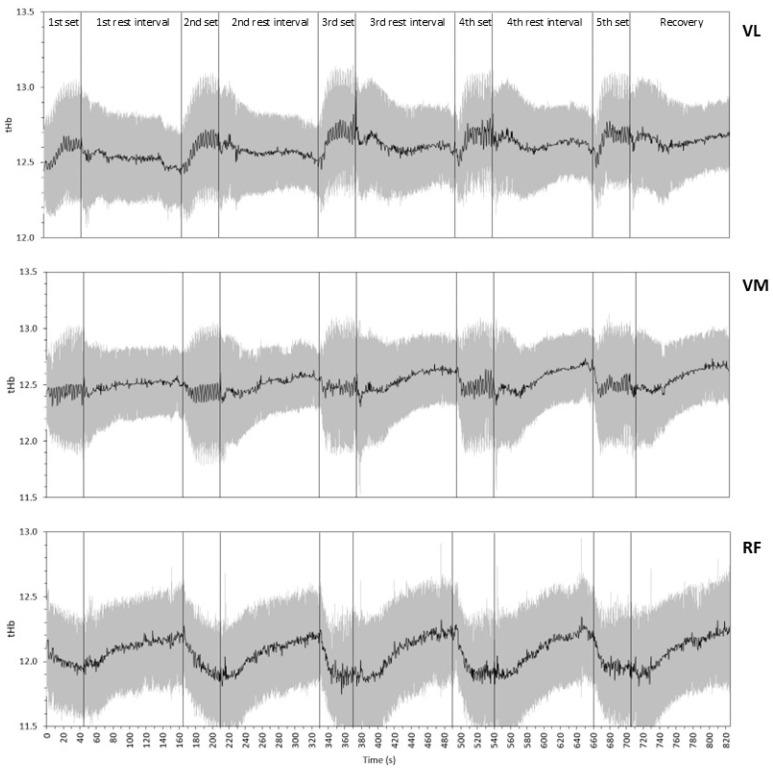
Total hemoglobin (tHb) kinetics in the vastus lateralis (VL), vastus medialis (VM), and rectus femoris (RF) during the testing protocol. Black lines indicate the mean and gray 393 whiskers indicate the SD.

**Figure 5 sports-12-00283-f005:**
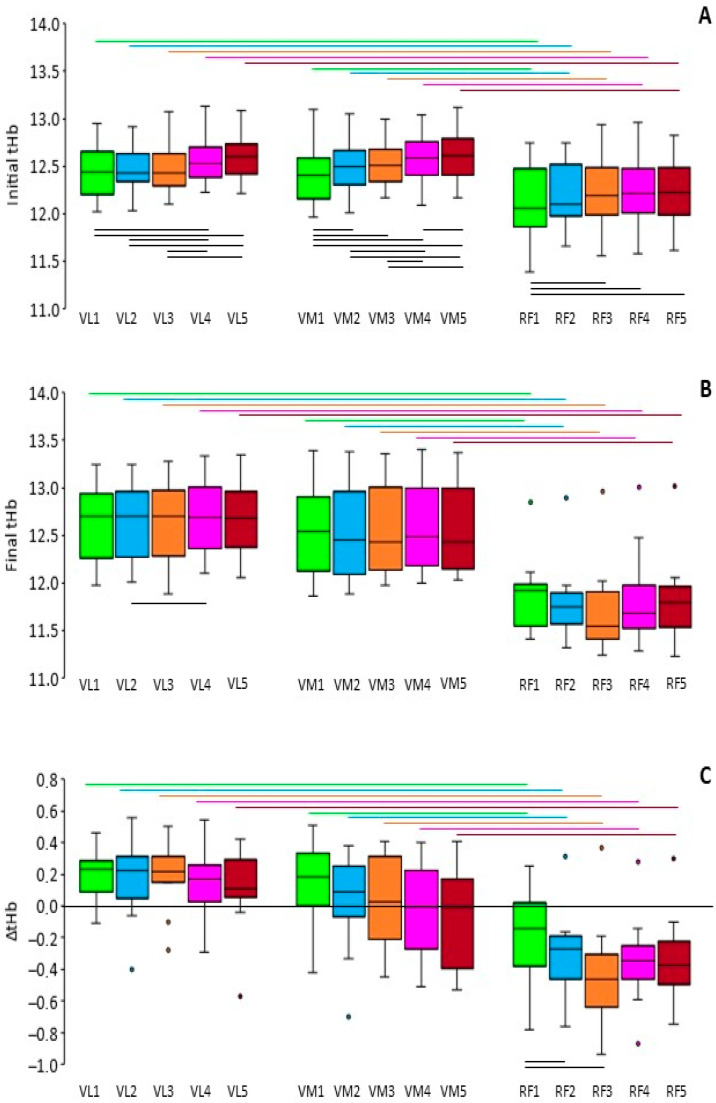
Boxplots of the initial values (**A**), final values (**B**), and change in the tHb (**C**) in the vastus lateralis (VL), vastus medialis (VM), and rectus femoris (RF) in each of the 5 sets. See Figure 3 for a description of each plot’s elements. Horizontal lines above the boxplots represent significant differences between the muscles in the same set, whereas horizontal lines below the boxplots represent significant differences between the sets in the same muscle (*p* ≤ 0.05).

**Figure 6 sports-12-00283-f006:**
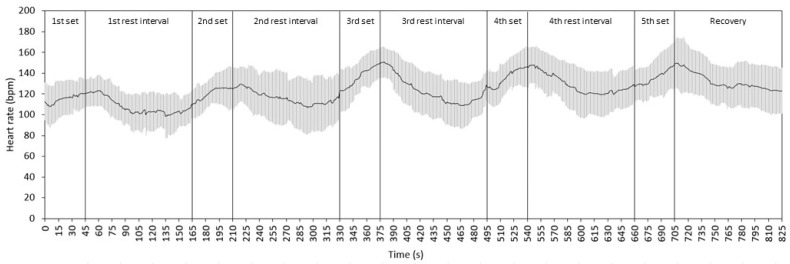
Heart rate kinetics during the testing protocol. Black lines indicate the mean and gray whiskers indicate the SD.

**Figure 7 sports-12-00283-f007:**
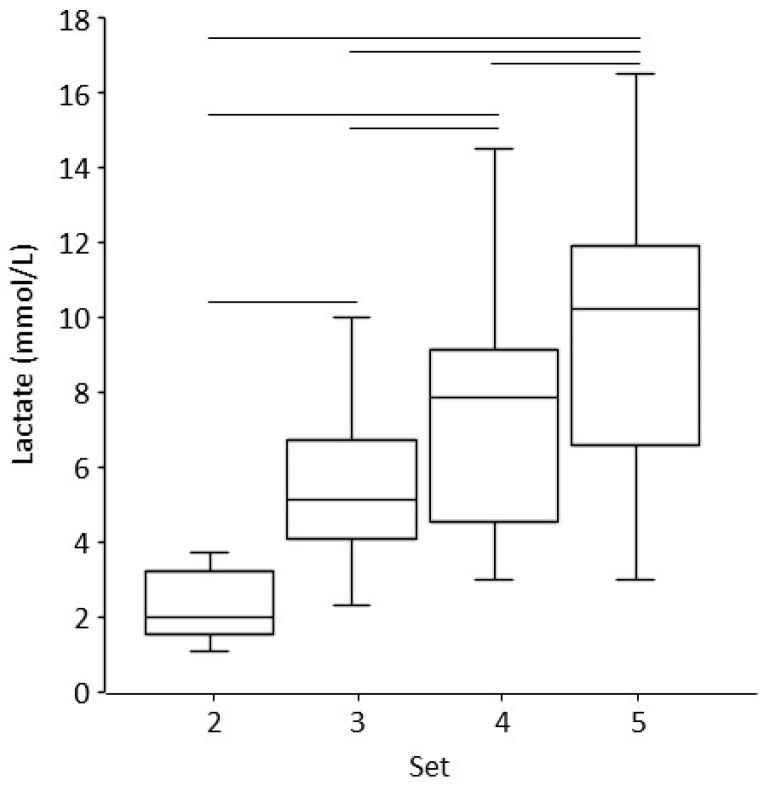
Boxplots of the blood lactate concentration after each set (except for the first one). See Figure 3 for a description of each plot’s elements. Horizontal lines above the boxplots represent significant differences between sets (*p* < 0.001).

## Data Availability

Our data are provided free of charge and can be accessed via the following link: https://doi.org/10.26255/heal.wraj-id6j, accessed on 15 July 2024.

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
