# Peer review of "Oxygenation Kinetics of Three Quadriceps Muscles During Squatting Exercise in Trained Men"

_sports, 2024, doi:10.3390/sports12100283_

Round 1
Reviewer 1 Report
Comments and Suggestions for Authors
The objective of the article is: The present study examined differences in oxygenation and blood supply between three quadriceps muscles during five squatting exercise sets of differing loads in trained men.
The authors are requested to address the following items:
1) Clearly describe the objective of the research at the end of the introduction section.
2) Provide examples of how muscle oxygenation determination contributes to the planning process of sports performance.
3) In the introduction, it is recommended to use concise and relatively short paragraphs, with each paragraph addressing the variables being analyzed (currently, only one paragraph is present).
4) Expand the introduction section, emphasizing the justification of the research and its utility in the sports training process.
5) Describe the type of research at the beginning of the methods section.
6) In the methods section, there are very long paragraphs that obscure the central idea and make reading difficult.
7) There is no evidence of a normality test for data distribution. Specify the normality test used and its results.
8) Certain data describing the sample characteristics (Section 3.1) are better suited for the methods section.
9) In the discussion section, describe the limitations of the research, with an emphasis on the sample size and its derived effects.
Comments on the Quality of English LanguageTo have the language evaluated by an English language specialist
Reviewer 2 Report
Comments and Suggestions for Authors
Intro
Could benefit from being a couple paragraphs
Don't use 1st person (we), It was hypothesized
L61: ....men (students at the....Science) participated....
Look for all instances of 1st person and change we to third person language
L92: is so-called correct? "the so-called" seems very conversational and thus not appropriate for scientific writing
Nice Figure
L130: I would remove the marginally significant information. You should err on the side of p value 0.05 or less as significant only
I didn't see any results that were marginally significant, so even more reason to delete. If I missed something that was marginally significant, it should be deleted and noted as insignificant
Reviewer 3 Report
Comments and Suggestions for Authors
The purpose of the study is not clear, and does not show a novel methodological approach to understanding the physiological action on training loads. Therefore, some parts of the methods and the discussion of the results are questionable.
Introduction
Since the late 1980s, NIRS has been used to study local muscle oxidative metabolism at rest and during different exercise modalities. Most studies have focused primarily on endurance or the ability to perform strength tasks and related determinants of muscle oxygenation (e.g., muscle oxidative capacity, oxygen delivery, and oxygen consumption). Of significance is the potential use of NIRS-related parameters as a metabolic biomarker in acute and chronic (e.g., training) responses of skeletal muscle function to exercise. Indeed, NIRS information can be used as a robust marker of skeletal muscle oxidative capacity. In healthy individuals, NIRS technology has shown good reproducibility regardless of the type of exercise.
What was the purpose of your study? Was it solely to demonstrate differences in the response to exercise of equal muscles of the same muscle group? This seems too little for a scientific article. It may have been worth referring to adaptations induced by exercise training after using a specific training protocol (endurance training, interval training with maximal/supramaximal intensity and annual planning). It is therefore necessary to put forward a research hypothesis, which you would like to verify here.
Materials and methods
What was the selection of groups for the study: random or purposeful?
For what purpose was the blood pressure measurement performed?
Results
Figures 3, 5 and 7 are illegible. Maybe it is worth changing them to a table. No F test value. No evaluation of results due to the studied effects: main and interactions. It seems that it is necessary to ask whether the factor of increased load caused a different muscle response. The ES values are small, add also the relative differences in percentages. It defines different variables but there is no evaluation of the relationships between them. Why?
Discussion
Why do you think that the response to load for all quadriceps muscles should be the same? This is a false assumption from the outset. These three muscles differ in function: The rectus femoris muscle is the only head of the quadriceps muscle that has its proximal attachment on the anterior inferior iliac spine of the hip bone, so it starts above the femur. This fact significantly affects its biomechanics, because unlike its companions, it affects the work of two joints - the hip and the knee. Studies focusing on measuring the moment of force of the rectus femoris have shown that, as a rule, the more the knee is bent, the shorter the lever arm. It is longest in flexion of about 45° of flexion in the knee joint, which makes the section 0° to 45° a certain exception to this rule. However, when the squat pattern (multi-joint exercise) was taken into account in the studies, it turned out that unlike other heads acting only on the knee joint - the rectus femoris acts on minimal leverage during this movement. This explains the small activation of the rectus femoris in multi-joint exercises and the small effects of squatting in hypertrophy training of this muscle. Due to the information presented, it is recommended to supplement multi-joint exercises developing the quadriceps with exercises of isolated knee extension in order to increase the activation of the rectus femoris muscle. Such exercises may include, for example, knee extensions in a sitting/lying position with an external load or reverse Nordic curl. The rectus femoris muscle turns out to be the most frequently injured head of the quadriceps muscle and is also associated with the highest frequency of re-injury (approx. 17%). The injury most often occurs during movements such as kicking or sprinting, i.e. movements characteristic for e.g. football. The same infomration are imortant to otherm heads of qudriceps muscles. In addition to the biomechanical specificity of the muscle itself, some researchers indicate the intramuscular tendon, which runs longitudinally along the front of the muscle, as one of the causes. In the introduction and in the discussion, the shoulder is an appropriate argument for the undertaken research. Since muscle functions differ, it was necessary to assess the factors that will affect the differences in the physiological response to load. This is not included in this article. It was possible to assess, for example, using EMG measurements. The conclusions are insufficient because they do not contribute anything new to science. It is necessary to indicate the advantages of these studies over others that have already been published on this topic.
Round 2
Reviewer 1 Report
Comments and Suggestions for Authors
The issued warnings have been resolved
Author Response
Thank you very much!
Reviewer 3 Report
Comments and Suggestions for Authors
It is therefore necessary to put forward a research hypothesis, which you would like to verify here. You have assumed that three muscles will differ in terms of muscle oxygenation and blood supply. Determine what biochemical and physiological reactions this exercise will cause. Since you measured blood lactate levels, heart rate and blood pressure, analyze the relationships between these variables - it is worth referring to the trend of these relationships (whether it is directly or inversely proportional).
Add the size effect to each p value.
You have already added information on muscle biomechanics, but it is also necessary to supplement information on differences in fiber composition between these muscles.
If you think your graphs are sufficient, remove the horizontal lines that are not labeled and it is known what they represent, and besides, they limit the range of the axis. In principle, in the text of the article you provide the value from the graphs anyway. Differences between muscles should be assessed as relative. However, you lack a reference level.
I do not understand what a serious level of hypoxia means. Why do you think it can serve as a hypertrophic stimulus? Could you please comment on the percentage of muscle fibers involved in this effort? Maybe then the assessed effect will be more understandable.
Round 3
Reviewer 3 Report
Comments and Suggestions for Authors
The required changes have been made.